Report

EMBO
*reports*

# Optogenetic inhibition of Delta reveals digital Notch signalling output during tissue differentiation

Ranjith Viswanathan[1,†,‡], Aleksandar Necakov[1,2,†], Mateusz Trylinski[3,4], Rohit Krishnan Harish[1,5], Daniel Krueger[1], Emilia Esposito[1], Francois Schweisguth[3], Pierre Neveu[6] (ID) & Stefano De Renzis[1,*] (ID)

## Abstract

Spatio-temporal regulation of signalling pathways plays a key role in generating diverse responses during the development of multi-cellular organisms. The role of signal dynamics in transferring signalling information *in vivo* is incompletely understood. Here, we employ genome engineering in *Drosophila melanogaster* to generate a functional optogenetic allele of the Notch ligand Delta (opto-Delta), which replaces both copies of the endogenous wild-type locus. Using clonal analysis, we show that optogenetic activation blocks Notch activation through *cis*-inhibition in signal-receiving cells. Signal perturbation in combination with quantitative analysis of a live transcriptional reporter of Notch pathway activity reveals differential tissue- and cell-scale regulatory modes. While at the tissue-level the duration of Notch signalling determines the probability with which a cellular response will occur, in individual cells Notch activation acts through a switch-like mechanism. Thus, time confers regulatory properties to Notch signalling that exhibit integrative digital behaviours during tissue differentiation.

**Keywords** morphogenesis; Notch signalling; optogenetics; signal dynamics; tissue differentiation

**Subject Categories** Development; Methods & Resources

## Introduction

Notch signalling is central to most developmental decision-making events in animals, and its misregulation is implicated in many diseases, including cancer [1]. Notch is a large transmembrane receptor activated by binding to its ligand Delta on the surface of neighbouring cells. Delta binding allows for the proteolytic cleavage of Notch and the subsequent release of the Notch intracellular domain (NICD), which translocates into the nucleus and regulates gene expression in a context-specific manner [2,3]. Lack of signal amplification between NICD generation and target gene activation, in addition to the fact that the Notch receptor cannot be re-used subsequent to an active signalling event, suggests a linear relationship in the transfer of information from the plasma membrane to the nucleus. Moreover, once cleaved, NICD molecules move into the nucleus within minutes [4]. While several studies have focused on the mechanisms controlling Notch activation by ligand endocytosis [5–9], knowledge about the activation of Notch targets by cleaved NICD and the dynamics involved remains limited [10–14]. More broadly, the role of signal dynamics in controlling developmental processes and tissue differentiation is only partially understood [15–19]. In this study, we employed optogenetics to modulate Notch signalling during *Drosophila* embryonic development in order to characterize its dynamic regulation and input–output relationship during tissue differentiation in real time. We focused on mesectoderm specification at the onset of gastrulation, which is defined by the expression of the transcription factor *sim* in two parallel single rows of cells flanking the mesoderm [20,21]. In agreement with the general paradigm of Notch signalling activation, Delta is internalized from the surface of mesodermal cells along with the Notch extracellular domain (NECD) in response to the expression of the ubiquitin ligase *neuralized* [8,9]. This results in Notch activation and *sim* expression specifically in the mesectoderm, as the mesoderm-specific transcription factor Snail represses *sim* expression in the mesoderm. While Delta internalization in the mesoderm initiates early during cellularization and proceeds in a uniform manner, *sim* expression starts only ~30 min later in what appears to be a gradual and random pattern of activation along the embryo antero-posterior axis (a-p) [20,21]. How the temporal dynamics of Delta internalization and Notch signalling activation relate to *sim* expression is unknown. Notch signalling might be required from the beginning of cellularization until *sim* transcription starts. Alternatively, there might be specific time intervals or a minimum threshold of NICD production required to activate *sim* expression. In more general

1  Developmental Biology Unit, European Molecular Biology Laboratory, Heidelberg, Germany
2  Department of Biological Science, Brock University, St. Catharines, ON, Canada
3  Institut Pasteur, UMR3738, CNRS, Paris, France
4  Sorbonne Université, Paris, France
5  Center for Regenerative Therapies Dresden, Technische Universität Dresden, Dresden, Germany
6  Cell Biology and Biophysics Unit, European Molecular Biology Laboratory, Heidelberg, Germany
   *Corresponding author. Tel: +49 6221 387 8109; Fax: +49 6221 387 8166; E-mail: derenzis@embl.de
   †These authors contributed equally to this work
   ‡Collaboration for a joint PhD degree between EMBL and Heidelberg University, Faculty of Biosciences

terms, these questions address principles linking signalling inputs to transcriptional outputs during tissue differentiation [18] and require methods to perturb endogenous signalling components acutely, while monitoring transcriptional responses. Here, we developed an optogenetic strategy to inhibit endogenous Delta activity with sub-minute temporal precision and simultaneously follow *sim* transcription in real time using the MS2-MCP system [22]. Using this approach, we show that while at the tissue-level Notch functions in an analog manner controlling both the timing and the frequency at which individual nuclei express *sim,* at the level of individual cells, Notch acts as a switch, with a minimum threshold of Notch activity determining whether *sim* is expressed or not. These results are consistent with a model in which Notch signalling performs digital time-integration during tissue differentiation.

## Results and Discussion

We generated a functional, endogenously tagged optogenetic allele of Delta (opto-Delta) by inserting a ϕC31 recombinase-landing site in the Delta locus, replacing a large part of the Delta coding sequence. The resulting heterozygous Delta mutant line served as an acceptor line allowing for the systematic screening of donor constructs carrying a cognate *attB* recombination sequence [23] (Fig 1A and B). opto-Delta rescue constructs were designed by identifying potential tagging sites through sequence conservation and linear motif analysis [24]. We identified an intramolecular polyalanine-rich region in the intracellular domain of Delta (aa 701), which was not conserved nor predicted to reside in a known folding domain. Insertion of an intramolecular GFP tag in this region resulted in fully viable Delta::GFP homozygous flies, with one copy of Delta::GFP capable of rescuing both a Delta loss-of-function mutant allele and a deficiency in *trans* (Fig EV1A–C). Potential opto-Delta constructs were designed based on the Cryptochrome 2 (CRY2)/CIB1 protein heterodimer-ization system from *Arabidopsis thaliana* [25]. Upon photo-activation with blue light, CRY2 interacts with its binding partner CIB1, but also oligomerizes when expressed on its own [26]. We reasoned that CRY2 might provide an optimal means by which to regulate endogenous Delta localization and function *in vivo* by, for example, interfering with the stoichiometry of endogenous Delta/Notch complexes, or altering the conformation of Delta molecules at the plasma membrane. We generated a series of constructs containing either a CRY2 tag alone (CRY2-PHR corresponding to residues 1–498) (Delta::CRY2), or a CRY2 tag fused to EGFP (Delta::CRY2::GFP) or tag-RFP (Delta::CRY2::RFP). Two additional constructs, containing either a CRY2-olig tag (a CRY2 variant with an increased propensity for blue light-induced oligomerization (Delta::CRY2-olig)) [27], or a CIBN tag (a CIB1 construct lacking the C-terminal nuclear targeting signal (Delta::CIBN)), were also produced (Fig 1B and C). After injection into the Delta acceptor landing line, individual fly stocks were screened for homozygous flies viable in the dark. While Delta::CRY2 and Delta::CIBN gave rise to fully viable and fertile homozygous flies, Delta::CRY2::GFP, Delta::CRY2::RFP and Delta::CRY2-olig homozygous flies were less frequent and displayed reduced fertility. This might be caused either by the concomitant presence of two relatively large tags (each ~35 kDa), which could

interfere with protein folding, or by increased dark activity (i.e. the excited state of a photoreceptor in the dark) of the CRY2-olig tag compared to CRY2.

When Delta::CRY2 flies were grown under normal light conditions, they developed severe Delta phenotypes characteristic of deficient Notch signalling during larval and pupal development (Fig 1D–Q). These defects include loss of wing vein margins (Fig 1D–G), a characteristic Delta-wing phenotype [28], increased density of sensory bristles in the adult notum (Fig 1H–K) [29,30] and altered eye morphology (Fig 1L–O) [31]. We also observed lack of embryo hatching in the light, suggestive of deficient Notch signalling during embryogenesis [32,33]. Inhibition of Notch signalling during embryogenesis results in an increase in neuronal density at the expense of ectodermal cell fate [34]. This phenotype is characterized by a lack of embryonic cuticle secretion, a process that is normally driven by ectodermal cells, while the development of larval structures such as the anal plate and mouth hooks remains unaffected [33]. Consistent with a loss of Notch signalling, Delta::CRY2 embryos exposed to continuous illumination under a normal dissecting microscope developed a Notch cuticle phenotype and did not hatch (Fig 1P–Q). Delta::CRY2 flies grown under light displayed more severe cuticle, eye and wing phenotypes than their Delta heterozygous counterparts, indicating a reduction of Delta activity > 50% (Fig EV1D–L).

To investigate the mechanisms underlying loss of Delta::CRY2 activity upon light exposure, we followed Delta dynamics and Notch signal activation in the early embryo. During early embryogenesis, Delta is uniformly localized at the plasma membrane and is internalized in the mesoderm at the onset of cellularization along with the extracellular domain of Notch [8,9]. This results in Notch activation and expression of the mesectoderm-specific transcription factor single minded (Sim), a direct Notch target, in a single row of cells flanking the ventral mesoderm [20,21]. Delta::CRY2::GFP, used as a *proxy* for Delta::CRY2, displayed normal plasma membrane localization in the first frame of acquisition with 488 nm light (< 1 s), but exhibited rapid segregation into plasma membrane clusters after only a few frames of acquisition ($t_{1/2}$ ~40 s) (Fig 2A–C, Movie EV1). Delta clusters were normally internalized in the mesoderm and persisted at the plasma membrane in the ectoderm (Fig 2D–G). Quantification of Delta plasma membrane levels revealed a ~70% decrease in the mesoderm compared to the ectoderm under both dark and light conditions (Fig EV2A–C). Furthermore, the plasma membrane-to-cytoplasmic (vesicles) ratio of Delta in the mesoderm did not change upon light exposure (Fig EV2D), and ~75% of Delta-positive vesicles colocalized with the early endosomal marker Rab5 in wild-type, Delta::CRY2 (dark) and Delta::CRY2 (light) embryos (Fig EV2E–H). Together, these data argue that Delta clustering did not interfere with the normal mechanisms driving Delta ubiquitination and internalization. Imaging endogenously tagged Notch::YFP in a Delta::CRY2 background revealed that Notch also formed plasma membrane clusters upon photo-activation in the ectoderm (Figs 2H and I, and EV3A). Antibody staining demonstrated Notch and Delta colocalization in ectodermal clusters, showing that opto-Delta clusters caused co-clustering of Notch and presumably its engagement (Fig EV3B–D). In mesodermal cells, staining against both Notch NECD and NICD showed reduced production of NECD endocytic vesicles (Fig 3A–D) and a ~30% increased retention of Notch at the plasma membrane upon photo-activation (Fig 3E–I).

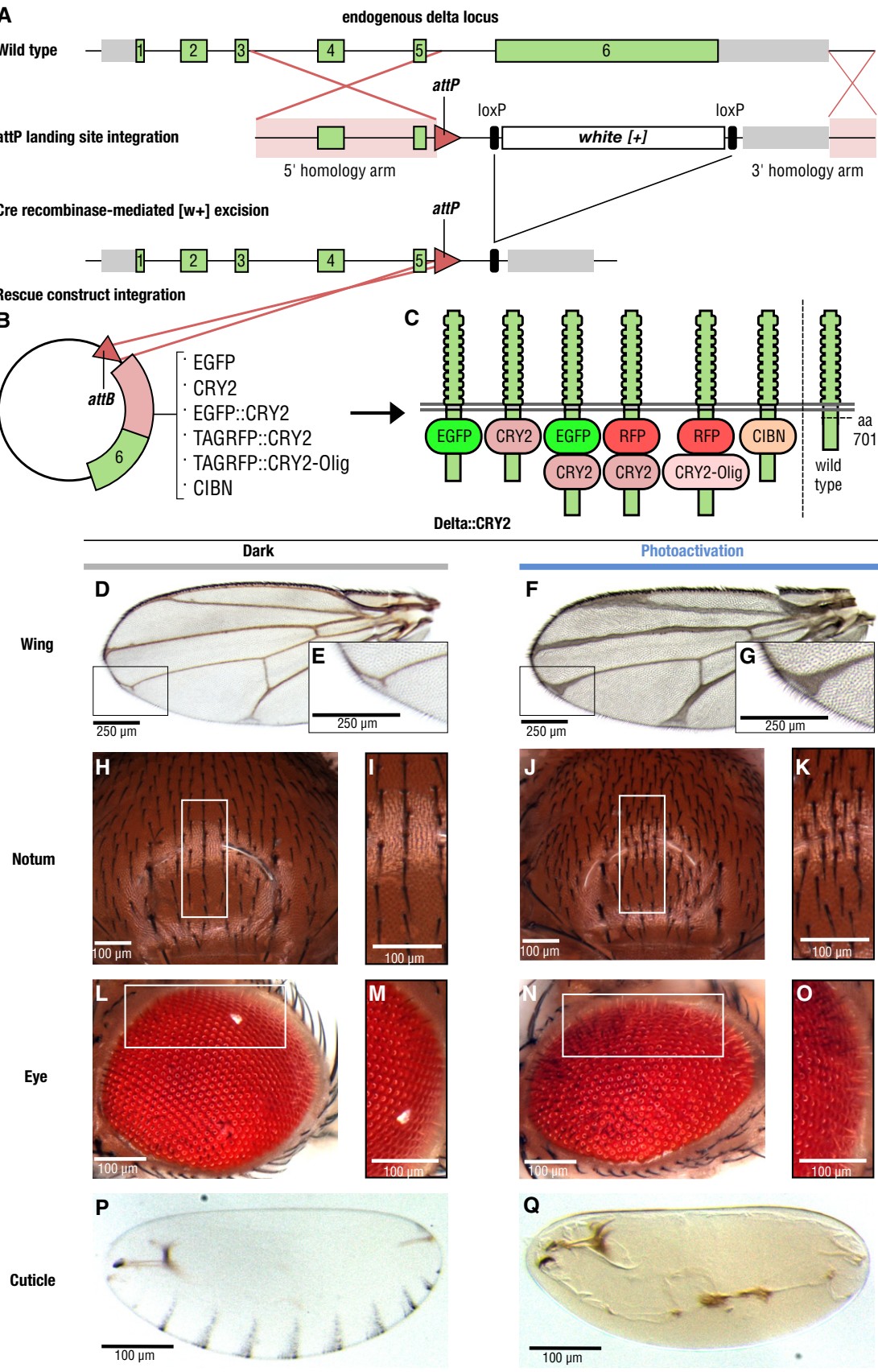

**Figure 1.**

**Figure 1.  Endogenous optogenetic tagging of Delta provides light-gated control of Notch signalling.**

A–C   Schematic illustrating the strategy used to generate a functional optogenetic-tagged allele of the Notch ligand Delta. (A) Gene structure of the Delta locus, green and grey bars represent exons and 5′/3′ UTRs, respectively. Exons 1–6 are shown in green, 5′ and 3′ in grey, homology arms are highlighted in pink, and recombination sites are indicated by red lines. Homologous recombination was used to replace a large portion of the Delta locus between the intron preceding exon 6 to a region downstream of the transcriptional stop site, through knock-in of an attP landing site and a loxP-flanked mini-white cassette. attP knock-in founder lines were identified as red-eyed transformants. Cre recombinase was subsequently used to remove the mini-white cassette, resulting in white-eyed flies. Delta-attP white-eyed founder lines were then transformed with attB rescue construct vectors carrying an attB recombinase binding site upstream of the genomic Delta sequence (B), along with incorporated sequences coding for a variety of tags inserted into an 11 amino acid polyalanine sequence in the intracellular domain of Delta (amino acid 701). (C) Position of the different tags with respect to the Delta protein sequence and its orientation in the plasma membrane.

D–Q   Flies homozygous for Delta::CRY2 are viable and fertile, and exhibit light-gated control of Notch signalling during development. Homozygous Delta::CRY2 flies raised in the dark exhibit only a mild Delta phenotype in the terminal tips of the wing (D, E—magnified view). Otherwise, these flies exhibit normal patterning and morphology in tissues including the notum (H, I—magnified view), the eye (L, M—magnified view) and the embryo (P). In contrast, Delta::CRY2 flies reared in the light exhibit Delta loss-of-function phenotypes, which include thickening of the wing veins (F, G—magnified view), an increase in microchaeta in the notum (J, K—magnified view), disorganization of the ommatidia (N, O—magnified view) and loss of denticle belt patterning in the embryo (Q). Scale bars, 250 μm in (D–G) and 100 μm in (H–Q).

These data suggest defective *trans*-endocytosis of Notch by Delta, potentially indicating that the signal-sending capacity of opto-Delta is compromised upon photo-activation. RNA *in situ* hybridization against *sim* confirmed a potent inhibition of Notch signalling (Fig 3J–K). However, given that some NECD vesicles still formed in the mesoderm upon photo-activation (Fig 3C inset) and that opto-Delta clusters in the ectoderm contained Notch (Fig EV3B–D), it is equally possible that opto-Delta inhibits signalling by blocking Notch through a mechanism involving *cis-repression* in receiving cells [35–37]. Distinguishing between *cis* versus *trans* inhibition is challenging as epithelial cells are closely packed together, making it difficult to perturb Delta specifically at the surface of individual adjacent signal-sending or signal-receiving cells. We turned to the pupal notum as an experimental model tissue owing to the fact that, in contrast to the early embryo, the notum is amenable to clonal analysis.

In the pupal notum, Notch signalling is necessary for proper bristle patterning through the process of lateral inhibition in multiple groups of cells called proneural clusters. The signal-sending cell becomes the sensory organ precursor (SOP), while the surrounding cells that receive the Notch signal are repressed from SOP fate and become epidermal instead (Fig 3M) [29]. To distinguish between signal-sending and signal-receiving defects induced by opto-Delta activation, clones of cells homozygous for opto-Delta in an otherwise wild-type background were generated using the FLP/FRT method [38]. FLP recombinase was expressed under the Ubx promoter, and opto-Delta homozygous clones were identified by the loss of nuclear-RFP expression (Fig 3L and N). The presence of an SOP marker made it possible for us to score for signal-sending and signal-receiving activity across boundaries (Fig 3N–Q). In a wild-type case, there is a 50% chance for a SOP cell forming on either side of a clone boundary [29]. Any significant shift from this percentage indicates a bias towards impaired signalling or receiving. Quantification of all the SOPs present at clone boundaries after 18 h of pupa formation (APF) showed that photo-activation resulted in ~90% (94.6 ± 2.6) of SOPs to be located inside the opto-Delta clones (Fig 3N and O). In the dark, this percentage was only slightly higher (62.5 ± 5.6) than the expected 50% value indicating random distribution (Figs 3O and EV4). Thus, upon photo-activation cells expressing opto-Delta have a higher chance to become SOPs compared to their neighbours suggesting reduced Notch receptor activity in these cells. In other words, opto-Delta activation inhibits the capability of cells to process Notch signalling in *cis*. The fact that boundary cells outside an opto-Delta clone always adopt a non-SOP fate demonstrates that opto-Delta cells are signalling competent. Taken together, we conclude that upon photo-activation opto-Delta preferentially affects signal receiving in *cis*, rather than signal activation in *trans*.

Having characterized the mechanisms through which opto-Delta activation inhibits Notch signalling upon photo-activation, we used it as a tool to characterize the input–output relationship of Notch signalling during mesectoderm specification. As readout

**Figure 2.  Delta::CRY2 undergoes rapid plasma membrane light-induced oligomerization.**

A   Schematic illustrating the oligomerization of opto-Delta molecules at the plasma membrane upon photo-activation with blue light.

B   Snapshots from confocal live imaging movies of Delta::GFP::CRY2 embryos (ectoderm stage 5) at the onset of photo-activation ($T_0$) or after 60 s and 120 s at 2-s intervals ($\lambda$ = 488 nm, 0.6 mW). Scale bar, 10 μm.

C   The kinetics of light-induced opto-Delta clustering were quantified in the ectoderm of Delta::EGFP::CRY2 embryos. Data were collected using confocal live imaging at 2-s intervals ($\lambda$ = 488 nm, 0.6 mW), and the relative intensity of Delta::GFP::CRY2 clusters in three embryos ($n$ = 3, 100 cells) was quantified over time using a custom-built image analysis pipeline developed in Cell profiler. Relative intensity of Delta clusters is shown over time (red line), with standard deviation between replicates highlighted in pink.

D–G   Immunostaining of Delta::CRY2 embryos using an anti-Delta antibody (green) at the end of cellularization in embryos fixed in the dark (D, E) or after batch photo-activation as described in Material and Methods (F, G). Panels display max. intensity $z$-projections of 10 slices at a $z$-interval of 0.7 μm. Magnified insets in (D, F) show the localization of Delta::CRY2 in the ectoderm in the dark (D) and after photo-activation (F). Embryos were co-stained with an anti-Tom antibody (E, G) in order to mark the ectoderm-mesoderm boundary. Delta-CRY2 protein clustered in the ectoderm and was normally internalized in the mesoderm. Scale bars, 10 μm and in insets 5 μm.

H, I   Notch clustering in response to light-induced clustering of Delta-CRY2. Sum-of-slice $z$-projection of 4 slices at 0.4-μm $z$-interval of cellularizing embryos expressing endogenously tagged Notch::YFP imaged using an argon laser ($\lambda$ = 514 nm) in a Delta::CRY2 heterozygous background at the onset of photo-activation ($T_0$), after 60 s and 120 s. Embryos were photo-activated once for 5 s before the 60-s and 120-s acquisition (stack size of $z$ = 10 μm, $\lambda$ = 488 nm, 0.6 mW). Scale bar, 10 μm. Schematic illustration of the redistribution of Notch into clusters (I).

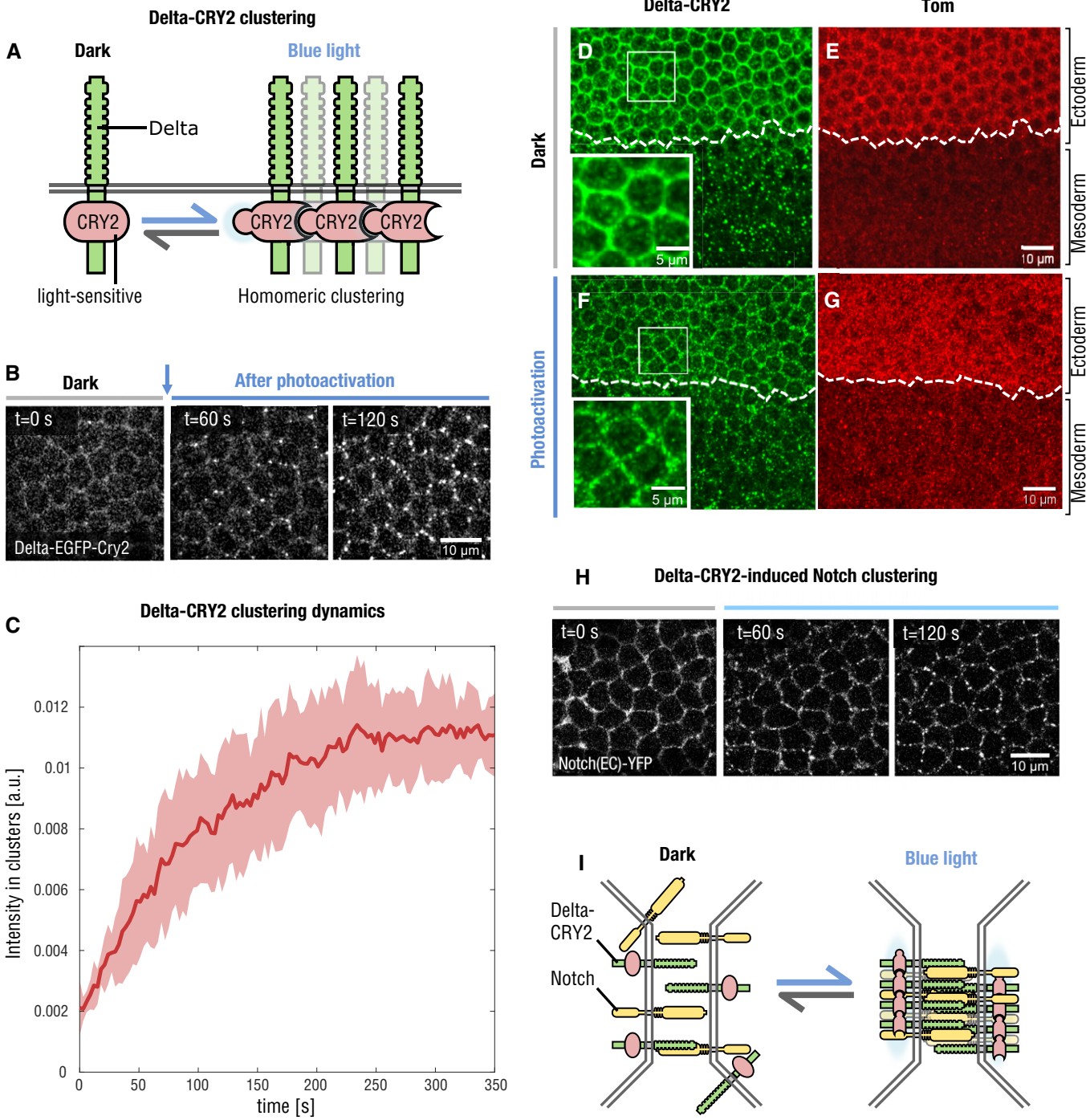

**Figure 2.**

for signalling, we constructed a transgene based on the MS2-MCP system [22] and used it as a live *trans*criptional reporter of *sim* expression in the mesectoderm. MCP::GFP interacts with the nascent *sim*-MS2 *trans*cripts to produce bright spots of fluorescence in the nuclei enabling an immediate quantitative measure of signalling activity (Fig 4A). In agreement with previous *in situ* hybridization data [20,21], *sim trans*cripts were detected approximately 30 min after the onset of cellularization (Fig 4B, Movie EV2). Thereafter, the number of nuclei expressing *sim* gradually

increased over time with an activation rate of ~0.2 min$^{-1}$ per µm until after 10 min almost all nuclei in the mesectoderm expressed *sim* (Fig 4B). The reason for this temporal gap in *sim* expression is unclear. Delta endocytosis and NECD trans-endocytosis begin within a few minutes after the onset of cellularization [8,9], and therefore, in principle, *sim* expression could occur almost concomitantly. During early cellularization, nuclei may not be competent to transcribe *sim* because additional factor/s necessary for its expression become available only at later stages of

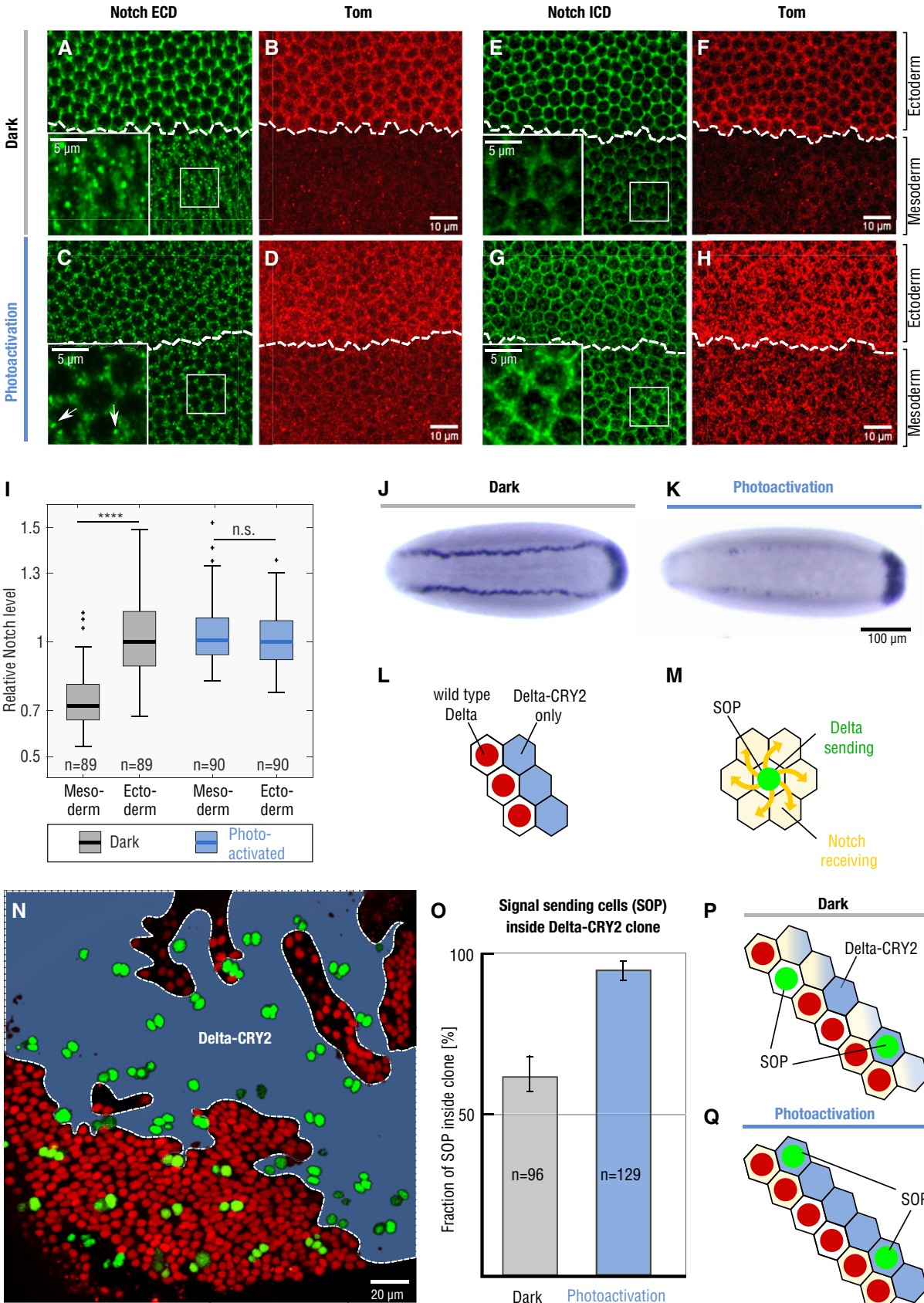

**Figure 3.**

**Figure 3. Optogenetic activation inhibits Notch in signal-receiving cells.**

A–I   Light-induced Delta::CRY2 clustering inhibits Notch processing in the mesoderm. Immunostaining for Notch Extracellular Domain (NECD) (A, C) or Notch Intracellular Domain (NICD) (E, G) at the end of cellularization in Delta::CRY2 embryos fixed in the dark or batch photo-activated as described in Material and Methods. Co-staining with an anti-Tom antibody (B, D, F, H) was used to mark the ectoderm–mesoderm boundary. In the dark, normal processing of Notch in the mesoderm results in the depletion of Notch (both NECD and NICD) from the plasma membrane, and accumulation of cytoplasmic NECD vesicles, magnified insets in (A, E). Photo-activation caused reduced number of NECD vesicles, arrows in magnified inset (C) and increased retention of Notch in the mesoderm, magnified insets in (C, G) as also demonstrated by the box plot in (I) showing quantification of NICD plasma membrane levels. NICD Interface intensity values were normalized to the median NICD value present in the ectoderm for each embryo ($N$ = 3 embryos; $n$ = number of interfaces with $n_{Dark}$ = 89 $n_{PA}$ = 90. ****$P$ < 0.0001, two-sample $t$-test. n.s. indicates not statistically significant differences). In the box plots, the solid horizontal line, bottom and the top edge of each box indicate the median, the 25th percentile and 75th percentile, respectively. Whiskers extend to the most extreme data point. "+" symbols indicate outliers. Scale bars, 10 μm, and in insets 5 μm.

J, K   *In situ* hybridization against *sim* in Delta::CRY2 embryos fixed in the dark (J) or after batch photo-activated started at the onset of cycle 14 as described in Material and Methods showing lack of *sim* transcription in the mesectodermal cells (K). Embryos are aligned anterior to the left and ventral side facing up. Scale bar, 100 μm.

L–N   Clonal analysis in the pupal notum to distinguish whether signal sending or signal receiving is impaired upon optogenetic activation. Schematic illustration depicting the clone boundary between Delta::CRY2 cells (nls-RFP negative) and WT cells (nls-RFP positive) (L). Selection of Sensory organ precursor cells (SOPs) in proneural clusters by the process of lateral inhibition, wherein signal-sending SOPs inhibit the surrounding signal-receiving cells from SOP fate (M). Notum at 18 h after puparium formation (N) showing Delta::CRY2 clones in blue and dashed lines depicting the clone borders where the number of SOPs (green) are scored. Scale bar, 20 μm.

O–Q   Analysis of SOP fate decisions across Delta-CRY2/wild-type clone borders. The bar plot (O) shows the percentage of SOPs located inside Delta::CRY2 clones out of the total number of SOPs scored at the border, in either dark or photo-activated conditions. In a wild-type case, there is a 50% chance for an SOP to be present on either side of the boundary, schematic in (P). In the dark, 62.5% of SOPs are present on the side of the Delta::CRY2 clone. This proportion significantly increased to 94.6% in photo-activated pupae, schematic (Q), suggesting a strong signal-receiving deficiency of Delta::CRY2 cells ($N$ = 3 pupae; $n_{SOP}$ = 96 (dark); $N$ = 5 pupae; $n_{SOP}$ = 129 (photo-activation)). $n_{SOP}$ signifies total number of SOPs at clone boundary. Error bars indicate standard deviation between different pupae. SOPs formed inside the Delta::CRY2 clones are able to inhibit their neighbouring cells on the wild-type side from adopting SOP fate, thus showing their signal-sending competence.

cellularization. Alternatively, Notch activation needs to reach a critical threshold to activate *sim*.

To distinguish between these two different modes of *sim* activation, opto-Delta embryos expressing the MCP-MS2 *sim* module were photo-activated either from the beginning of cellularization (0 min), or after 5, 10, 15, 20 min, or 25 min of signaling (Fig 4C and D, and Movies EV2–EV7). The result of this experiment showed that the longer the photo-activation time, the slower the activation rate of *sim* expression (Figs 4E and F, and EV5A), with 20 min of signalling representing the minimum time necessary for all nuclei to have an equal probability to turn on *sim* with the same activation rate as control embryos (Fig 4F). Furthermore, all photo-activation conditions except for 25 min caused a significant delay (~10 min) in the

onset of *sim* expression (Fig 4G) and reducing Notch protein levels by half (Notch heterozygous embryos) caused a similar significant delay in the onset of *sim* expression (Fig EV5B and C). Together, these results suggest that photo-activation causes changes in the activation kinetics of *sim* expression, which are compatible with reduced levels of Notch activity.

The graded tissue-level pattern of *sim* activation was not reflected at the level of individual nuclei. In agreement with a recent study of *sim* enhancer regulation by NICD [14], *sim* nuclear dots were either absent or present and their intensity remained relatively stable (fluctuating around the mean) for the entire duration of cellularization (Fig 4H), as revealed also by quantitative fluorescent *in situ* hybridization of nascent

**Figure 4. Optogenetic inhibition of Notch signalling reveals time-integrated digital regulation of target gene expression.**

A   Schematic illustrating the use of light-induced Delta::CRY2 clustering in combination with the MCP-MS2 live transcriptional reporter of *sim* expression.

B   Graph illustrating the gradual onset of *sim* transcription in mesectodermal nuclei during the time-course of nuclear cycle 14 in embryos expressing MCP-GFP and *sim*-MS2. Following a delay of approx. 30 min, *sim* spots began to appear stochastically in the tissue and increased the rate of 0.2 spots min$^{-1}$. The rate is calculated by fitting the averaged curve as shown in Fig EV5. Solid line represents the mean, and shaded area represents the standard error of mean, $N$ = 3 embryos.

C   Schematic representing the optogenetic experiments presented in (D–H) where different periods of signalling were allowed from the onset of cycle 14 (0/5/10/15/20/25 min) prior to beginning of photo-activation in embryos expressing Delta::CRY2, MCP-GFP and *sim*-MS2.

D   Segmentation of MCP-GFP-positive nuclear spots from confocal movies at the final time-points acquired prior to ventral furrow formation in either control (non-Delta::CRY2) embryos, which were photo-activated for 60 min or Delta::CRY2 embryos in which signalling was allowed for either 20 min, 5 min, 0 min. Scale bar, 20 μm.

E   Plot depicting the percentage of *sim* transcribing nuclei over time (with respect to control embryos) during the different time-periods of photo-activation. The onset of ventral furrow formation is considered as $t$ = 0 min. Solid lines represent the mean, and shaded areas represent the standard error of mean. $N$ = 5 embryos for control, 0 min sig. and 15 min sig.; $N$ = 6 embryos for 10 min sig. and 20 min. sig.; $N$ = 7 embryos for 5 min. sig. and 25 min. sig.

F   Bar plot showing the gradual increase in the rate at which mesectodermal nuclei start to turn on *sim* (*sim* spot density/min) as the duration of signalling increases from 0 min to 25 min as depicted in (C). The dataset used for quantification is the same as that used in (E). *$P$ ≤ 0.05, **$P$ ≤ 0.01, ***$P$ ≤ 0.001, n.s. represents no statistically significant difference in a 2-sample $t$-test. Error bars denote standard error of mean.

G   Bar plot showing that onset *sim* transcription is significantly delayed compared to control embryos as the duration of signalling is increased from 0 min to 20 min. After 25 min of signalling, there is no significant delay. The dataset used for quantification is the same as that used in (E). *$P$ ≤ 0.05, **$P$ ≤ 0.01, n.s. represents no statistically significant difference in a 2-sample $t$-test. Error bars denote standard error of mean.

H   Graph showing that gradually increasing the time-period of active signalling from 0 min to 25 min does not change the mean spot intensity of *sim* transcripts compared to controls. There was only a 20% variation in spot intensity over time, a value that was smaller than the ~75% variation in spot intensity for a given time point. The dataset used for quantification is the same as that used in (E). Shaded areas represent the interquartile range.

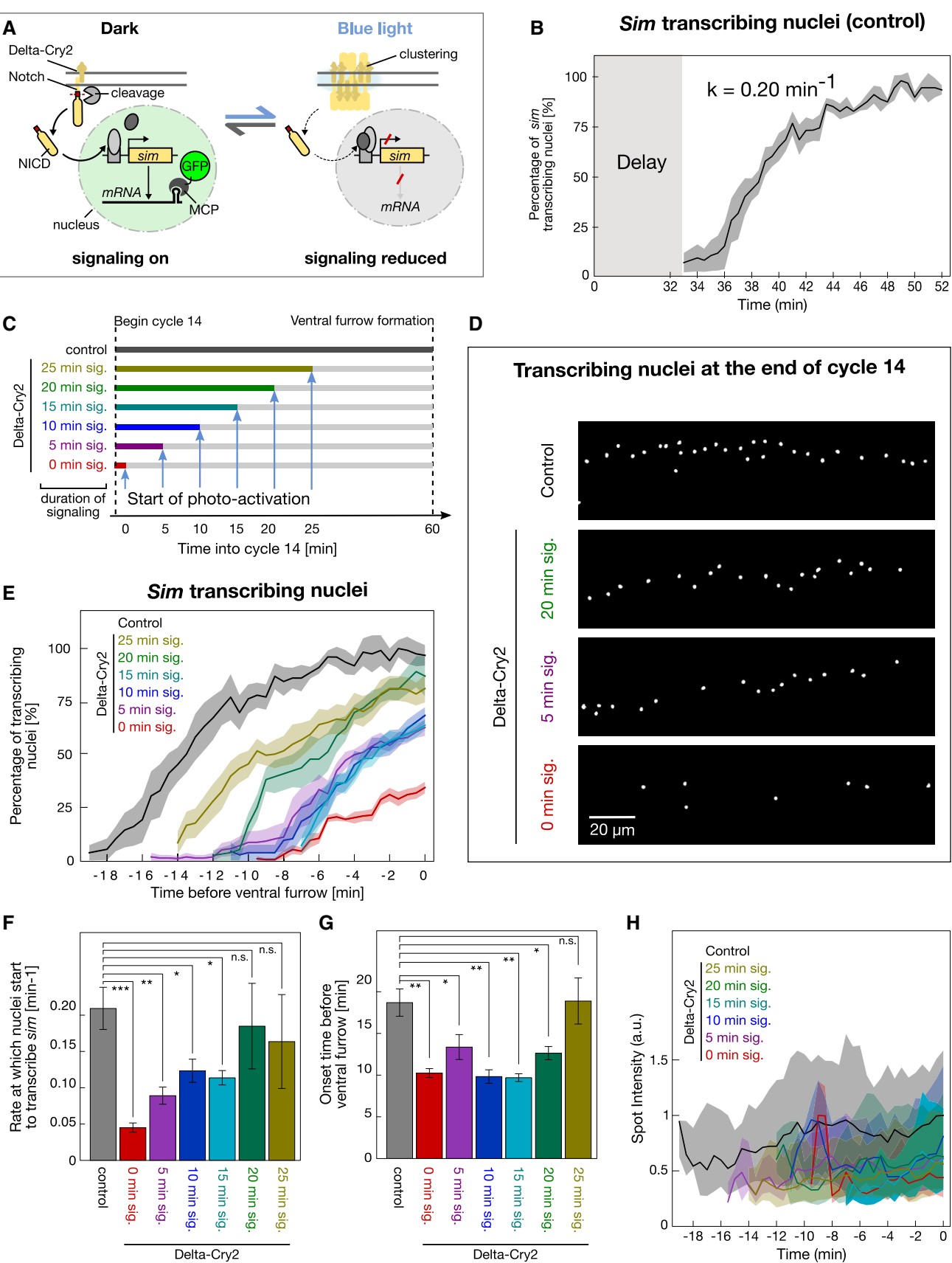

**Figure 4.**

transcripts (Fig EV5D–I). These results suggest that a cumulative amount of Notch activation determines the competence of mesectodermal nuclei to express *sim*, and that the time-lag between the beginning of cellularization and *sim* expression is Notch dependent. Increasing the time of Notch activity from 5 min to 10 min to 20 min had no significant impact on the mean nuclear dot intensity of sim transcripts from the time-point at which they first became detectable until the endpoint of imaging. Thus, in individual nuclei *sim* expression is controlled by a switch-like digital Notch-dependent mechanism.

Taken together, our results indicate two distinct relationships between Notch input and *sim* output. At the level of individual cells, Notch acts in a switch-like manner, with a minimum threshold of Notch activity determining whether *sim* is expressed or not. Above a certain threshold, *sim* expression is insensitive to changes in Notch activity. At the tissue level, Notch exhibits an analog-like regulatory mode with the level of its activity controlling both the timing and the frequency at which individual nuclei express *sim*. This regulatory mode is a critical factor in ensuring that all mesectodermal nuclei express *sim* by the onset of gastrulation and support a model in which the Notch receptor is an integrator of (noisy) analog signals that generates a digital switch-like behaviour [39] at the level of target gene expression during tissue differentiation. The physiological implications of this regulatory mode can be understood in the context of thresholded non-linear responses, which are known to confer robustness of signalling outputs to fluctuation in inputs [40]. It is tempting to speculate that time-integrated analog-to-digital conversion of Notch signalling may function to minimize spurious target gene expression resulting from transient cell–cell contacts during highly dynamic morphogenetic movements.

# Materials and Methods

### Endogenous targeting of the delta locus

Homologous recombination was used to target the endogenous Delta locus following a procedure described [41]. Briefly, a large portion of the Delta locus between the intron preceding exon 6 to a region downstream of the transcriptional stop site was replaced with a targeting cassette through knock-in of an attP landing site and a loxP-flanked mini-white cassette. As a first step, a transgenic donor line containing the attP-containing targeting cassette on chromosome 2 was established. This targeting cassette carried flanking homology arms including sequences extending ~5.5 kb upstream of the intron immediately upstream of exon 6 (Left Homology arm), and ~3 kb downstream of the transcriptional stop site (Right Homology arm). The transcriptional termination site along with sites of low sequence conservation at insertion sites was identified using the UCSC Genome browser in order to mitigate any potential detrimental mismatches arising as a consequence of errant recombination events. The Delta-attP targeting cassette was mobilized through a combination of FLP/FRT and Isc1 genomic excision/cleavage, resulting in a linear, double stranded, extra-chromosomal targeting construct. Heterozygous Delta-attP founder lines were identified as red-eyed transformants that exhibited an obvious Delta-wing phenotype resulting from the haploinsufficiency of the Delta-attP loss-of-function allele. Candidate transformants were fully characterized and validated by PCR and Sanger sequencing, resulting in the identification of a *bona fide* Dealt-attP founder line. The mini-white was subsequently excised from the Delta-attP locus through Cre/Lox-mediated recombination. This white-eyed Delta-attP founder line was then used as a substrate to generate Delta::EGFP, Delta::CRY2, Delta::EGFP::CRY2, Delta::TagRFP::CRY2, Delta::TagRFP::CRY2 Olig fly lines through Φ C31-mediated attp/attB recombination. An attB vector containing the entire Delta genomic sequence that was removed from upstream of exon 6 to downstream of the transcriptional stop site was produced for each of these rescue constructs, which were engineered to incorporate the coding sequences of their respective tag as an intramolecular fusion inserted between amino acids 701 and 702 of Delta protein. Transformants were identified by their red-eye colour, and recue of the Delta haploinsufficient wing phenotype, and were subsequently verified by PCR and Sanger sequencing.

### Live imaging and optogenetics

Cages with flies expressing Delta-CRY2 were maintained in the dark. Late stage 4 (Cycle 13) embryos were selected under halocarbon oil and mounted using a standard stereomicroscope with a red-emitting LED as the transmission light source. Mounting for live imaging was carried out as follows: embryos were dechorionated with 100% sodium hypochlorite for 2 min, rinsed with water and mounted immersed in PBS onto a 35-mm glass-bottom dish (MatTek corporation). Embryos were then positioned with their ventro-lateral side facing the cover-slip. Photo-activation and acquisition of movies were done with a spinning disc Ultraview VoX system (PerkinElmer) using a 40×/NA 1.3 oil immersion objective (Zeiss). Live imaging was performed at 25°C. Bright-field illumination was filtered using a Deep Amber lighting filter (Cabledelight, Ltd.) in order to locate the embryos. The microscope was controlled using Velocity software (PerkinElmer).

For imaging the kinetics of Delta-CRY2 clustering live, embryos heterozygous for Delta::GFP-CRY2 embryos were imaged using fluorescence live imaging (PerkinElmer Vox, Spinning Disc Confocal, 63× Oil immersion objective). A single plane, 3 μm below the apical surface, was imaged ($\lambda$ = 488 nm, laser power out of the objective = 0.6 mW) over six minutes at two-second intervals in three individual embryos ($n$ = 100 cells in total).

To image the clustering of Notch Extracellular-YFP in a Delta-CRY2 heterozygous background, at $t$ = 0 s, a pre-activation stack was acquired using a 514 nm laser, followed by an immediate cycle of photo-activation (5 s, $\lambda$ = 488 nm, laser power out of the objective = 0.6 mW). Thereafter, a post-activation stack was acquired using the 514 nm laser at $t$ = 60 s. The above cycle was repeated till $t$ = 120 s. For imaging the clustering of Notch Extracellular-YFP in a Delta-CRY2 homozygous background, there was no time delay between the pre-activation, photo-activation and post-activation stacks. All stacks were 10 μm starting from the most apical plane of the embryo surface with a $z$-interval of 0.4 μm.

For clonal analysis, Delta-CRY2 pupae were collected at puparium formation and either photo-activated under a white lamp source (20 W) or placed in the dark at 25°C until 16.5 h after puparium formation (APF) for live imaging as described previously [42]. Imaging was done with a Zeiss LSM 780 confocal microscope with a 63×/NA 1.4 oil immersion objective (Carl Zeiss).

For *sim* expression, virgin females of the line w; MCP (No NLS)-GFP/CyO; Delta-CRY2/" were crossed to males expressing the *sim*-MS2 reporter. Resulting embryos were mounted for imaging and photo-activated after allowing incremental time intervals (0/5/10/15/20/25 min) of signalling from the onset of cycle 14. Photo-activation was carried out by illumination with a 488 nm laser line (200 ms/slice, laser power out of the objective = 0.6 mW) every minute by gradually increasing the stack size from 20 to 25 to 30 μm (z-interval of 0.8 μm) from the apical surface as cellularization proceeded. *sim* nuclear spots were recorded also using a 488 nm laser at a time resolution of 30 s and a stack size of 25 μm (z-interval of 0.4 μm) until the onset of ventral furrow formation. For controls, female virgins expressing MCP-GFP (without Delta-CRY2) were crossed to males expressing the *sim*-MS2 reporter and photo-activated using the same protocol described above from the onset of cycle 14. For imaging *sim* expression in Notch heterozygous embryos, female virgins expressing one copy of the mutant Notch[55e11] allele and MCP-GFP were crossed to males expressing *sim*-MS2. Image acquisition in the resulting embryos and corresponding controls (without Notch[55e11]) was started as soon as *sim* spots began to appear, without any prior photo-activation.

Imaging conditions for visualizing nascent *sim* transcripts by fluorescent *in situ* hybridization in wild-type (w1118) embryos were identical to that used for the MS2-MCP system, except that a 63× Oil immersion objective was used instead of 40×.

## Optogenetic inhibition of Notch signalling during development and analysis of mutant phenotypes

Flies homozygous for Delta::CRY2, which are viable and fertile in the dark, were reared on standard German food (Bloomington *Drosophila* stock centre recipe) in standard acrylic fly vials and exposed to ambient light from larval stages until eclosion. These adults, along with their counterparts reared in the dark, were imaged on a Zeiss stereo-dissection microscope in order to characterize variations in wing, notum and eye morphology. For embryonic analysis, homozygous Delta::CRY2 embryos were collected for two hours on apple-juice agar plates in acrylic cages maintained either in the dark, or exposed to ambient light for 24 h at 25°C. Cuticle preparations of collected embryos were made using Hoyer's medium and imaged on a Zeiss stereo-dissection microscope. For Delta-CRY2 adult nota and eyes, the following protocol was used: pupae were collected at the puparium formation stage and illuminated under a white lamp source (20 W) or placed in the dark at 25°C until they hatched. Adults were frozen at −20°C for 15 min and then immediately prepared for bright-field imaging using a Zeiss AxioZoom V16 macroscope.

## Image analysis

To quantify Delta::GFP::CRY2 clustering kinetics, a data pipeline was developed in Cell Profiler to automatically identify and segment individual Delta::GFP::CRY2 clusters in a series of individual confocal images. Segmentation serves to first map the positional coordinates for each cluster, allowing for automatic quantification of the number of clusters in each image. Delta protein abundance in individual clusters was then automatically quantified by integrating fluorescence intensity across the total area of each cluster as identified

through segmentation. Data normalization was performed through the following formula: (Total Number of Clusters × Mean Intensity of Clusters)÷Total Intensity of Image. This approach to normalization provides a quantitative measure of the total amount of Delta protein (Delta::GFP:CRY2 locked into clusters), and the resulting data were plotted as the relative intensity of Delta clusters over time plus or minus standard deviation over time. This image analysis pipeline automatically identifies and segments individual Delta-GFP-CRY2 clusters in a series of individual images collected using spinning disc confocal microscopy. Segmentation serves to first map the positional coordinates for each cluster, allowing for automatic quantification of the number of clusters in each image. Delta protein abundance in individual clusters is then automatically quantified by integrating fluorescence intensity across the total area of each cluster as identified through segmentation.

For the quantification of Notch protein levels at the plasma membrane, confocal images were processed using Fiji and analysed using MATLAB-R2017b (MathWorks). A sum-of-slice projection of 10 focal planes was used to segment line profiles with a line width of 7 pixels, which were drawn across approximately 30 interfaces in both ectoderm and mesoderm of each embryo, and the corresponding intensity profiles were extracted in Fiji. These data were then input into a customized MATLAB pipeline. Peak values in the intensity profile were identified using the findpeaks function in MATLAB, and the integrated intensity across individual interfaces was calculated by choosing an interval of 0.78 μm centred at the peak.

For the clonal analysis, the total number of sensory organ precursor cells (SOPs, marked by neur-iRFPnls) at the border of wild-type and Delta-CRY2 clones were counted manually and scored for their presence on the side of either the wild-type or Delta-CRY2 homozygous clone. The number of SOPs on the side of the Delta-CRY2 clone was then plotted as a percentage of total number of SOPs at the border in both the photo-activated and dark conditions.

For the analysis and quantification of *sim* expression using the MS2-MCP system, the numbers of spots and associated intensity were quantified under different photo-activation regimes. In order to remove noise, individual images from stacks were median-filtered with a 3 × 3 pixel neighbourhood. A mean-filter over a 100 × 100 pixel neighbourhood was used to determine the background and subtracted to the median-filtered image. We then computed the maximum projection. We subtracted to the maximum projection image its median-filtered image with a 30 × 30 pixel neighbourhood. The resulting image was Gaussian-filtered with a standard deviation of 3 pixels (corresponding to the size of a diffraction-limited spot under our imaging conditions). The segmented image was built from the zero crossings of the Laplacian of the filtered maximum projection. Segmentation of background was rejected based on blob size and intensity. The final segmented image was manually curated. The segmented image was used as mask to extract the intensity of individual *sim* puncta.

For the quantification of *sim* expression using FISH, nascent *sim* transcribing spots in the nucleus were segmented, and their intensities were quantified in fixed embryos using a similar methodology as described above. In nuclei containing two visible transcriptional foci, both of them were used for quantification. The staging of embryos as early, middle and late during the course of

cellularization was done as follows: the early group was identified as having detectable levels of *sim* expression, and the cellularization furrow was measured to be at least 12 μm from the vitelline membrane but still positioned apical to the base of the cortical nuclei. The middle group was identified as having detectable levels of *sim* expression, and the cellularization furrow was positioned anywhere between the base of the nuclei and up to three microns below. The late group was identified as having detectable levels of *sim* expression, and the cellularization furrow was positioned at least 5 μm below the base of the nuclei but prior to ventral furrow formation.

To quantify Delta protein content at the plasma membrane, cell outlines were segmented based on the E-cadherin signal co-stained with Delta using Embryo Development Geometry Explorer (EDGE) software [43] (https://github.com/mgelbart/embryo-development-geometry-explorer). Image stacks spanning 3.5 μm were projected in 2-D using sum-of-slices. The membrane-bound Delta signal was measured in a 0.3-μm-thick region lining the segmented membrane, and for each embryo, the mean intensity value of the ectoderm and of the mesoderm was calculated. To quantify the membrane-to-cytoplasmic ratio of the Delta signal in the mesoderm, the central (cytoplasmic) signal was derived from the inverted mask of the segmented membrane and divided by the Delta signal measured at the surrounding cell edges. An average membrane-to-cytoplasmic ratio was calculated from ~40 cells per embryo.

To analyse the colocalization of Delta and Rab5 in the mesoderm, vesicles were detected in a 7-μm-spanning image stack using the Fiji "Spots colocalization" plugin ComDet v.0.4.1 (https://github.com/ekatrukha/ComDet). Identified Delta particles were binary classified for an overlap with Rab5 particles in at least 1 pixel. The percentage of Rab5-positive Delta vesicles were calculated per embryo.

### Estimation of activation rates

The temporal evolution of transcription spot numbers was adjusted by $N_{max} (1-\exp(-k (t-t_0))$ where $N_{max}$ is the maximum number of spots that can be activated (i.e. the number of nuclei in the mesectoderm), $k$ is the activation rate, and $t_0$ is the delay to activation. $N_{max}$ was taken as constant, and only two parameters $k$ and $t_0$ were adjusted for each embryo.

### sim-MS2 reporter line

To generate the *sim*-MS2 line, the *sim* enhancer and promoter region shown below was fused to 24 MS2 stem loop repeats. Cloning was performed identical to the *eve* > MS2 construct described previously [44].

#### Sim enhancer (664 bp)

AGCAGAGCTCTTATCGTTGTGGCCCCGGCATATGTTACGCACATTT
ACAGCGTATGGCGATTTTCCGCTTTCCACGGCCACGGCCACAGCT
TCCCACCTGATAGGACAGCTCGGCAATGTGTGGGAATCGCAGTGA
GGTGCCGGTAGGAGTGGCAGGTAAGCCTGGCCGCCTCGCAAGTTT
CTCACACTTCCAGGACATGTGCTGCTTTTTTGGCCGTTTTTCCCCG
ACTGGTTATCAATTGGCCGATTGGAAATTCCCCGATGGCGATGCG
CTAGCGTGAGAACATGAGCTGCGAGCATCGGGTTTTTAGCATATC
CATACCTGTGGCTCGTCCTGATGGGAAGCGAGAAGCAGCAGGATC
GGATGTAGGATGCAGGATATAGGGTATAGGCGCTGTTGCGCCTCA
CCCGCAACACCCACATTAGCATCGGACCAGCGTCCAGTGTCCTGT
TAATTGCTTTATGGACTCTCCACTTTCCGCTGCGTGGGAATCTTTG
CTCATCCTACCTGTTTCCATGCCACACCAACCCATTCCCACAGCAT
TGTCCTCCTTATGTGAAACTCTCTAGTTCAAGTTCAGTGTGAATAT
TTGTGTTGACTTTATTTTTAAACTTTTGGCCATTTGTTTTCAGTTT
CTGTTTGCCTGTAACCAGATTAAGGTC

#### Sim promoter (181 bp)

AATCCAGTGCAGCCAATGGCAGGTTGTTTTCTCAGGATCAGGTAA
CAGATCCTTTTCGGGATCAGTTGGGAAACTGTTAAAAGTGCTTGT
GCCGCTGGAAAGCGGCTCAGTTGCAAACAGGTGATTGCAGGGATA
TGAGCAAGTGCTGAGAAGGTGCTCGCAACAGTCTCAAAGCAGGA
TC

### Immunostaining and *in situ* hybridization

Delta-CRY2 embryos enriched in stage 4 and 5 were collected, dechorionated in bleach for 2 min, covered in PBS and photo-activated with the following illumination protocol: 2 min under blue light from a fluorescence lamp source (Olympus X-Cite 120 W, 470/40 bandpass filter. Light power out of the objective = 1.8 mW) 5 min under a white light source (HAL-100 Zeiss) and 3 min with no illumination. This protocol was repeated six times to ensure a total photo-activation period of 1 h. Thereafter, embryos were immediately fixed in 4% Paraformaldehyde (Electron Microscopy Sciences) and Heptane (Sigma) for 20 min after which they were devitellinized and stored in methanol at −20°C.

For Immunostaining, fixed embryos were blocked in 10% bovine serum albumin (BSA) in PBS 0.1% Triton-X (Sigma) for 1 h, following which they were incubated overnight in the primary antibody diluted in PBS containing 5% BSA and 0.1% Triton-X. The embryos were then washed and incubated in the secondary antibody for 45 min at room temperature. After another round of washing, embryos were mounted on glass slides using aqua-poly/mount (Polysciences Inc.). For the non-photo-activated control, embryos were collected under Deep Amber filtered light (as mentioned previously) and fixed in the dark prior to immunostaining. Images were acquired with a Zeiss LSM 780 NLO confocal microscope with a 63×/NA 1.2 water immersion objective (Carl Zeiss).

The following antibodies were used: mouse anti-extracellular Notch EGF repeats-12-20 (Developmental Studies Hybridoma bank, DSHB) (1:20 dilution); Mouse anti-Delta (DSHB) C594.9B (1:100 dilution), Mouse anti-intracellular Notch C17.9C6 (1:20 dilution) (DSHB), Rat anti-Tom (Home-made, 0.5 μg/μl) (1:100 dilution), Rabbit anti-E-Cadherin sc-33743 (1:100 dilution), Rabbit anti-GFP ab6556 (1:1,000 dilution) and Sheep anti-DIG 11333089001 Roche (1:500 dilution). Secondary antibodies used were anti-mouse Alexa 488, anti-mouse Alexa 647, anti-rat Alexa 647, anti-rabbit Alexa 488 and anti-Sheep Alexa 488; the dilution was 1:500.

Protocol followed for *in situ* hybridization was as previously described [9]. For fluorescent *in situ* hybridization, 1% skim milk was used in the blocking solution instead of 5% BSA and DAPI was used as a nuclear counterstain in order to facilitate the discrimination of nuclear transcriptional foci.

## Fly strains and genetics

All stocks were maintained by standard methods at 22°C, unless otherwise specified, and lines carrying the optogenetic CRY2 tag were stored in the dark as previously described [45].

In order to induce clones in the pupal notum:

w; Ubx-FLP neur-iRFP670nls/+; FRT82B ubi-RFPnls/FRT82B Delta::CRY2.

Clones were detected by the loss of nuclear-RFP.

## Fly stocks

w;; Delta::EGFP/". Endogenous Delta tagged intracellularly with EGFP

w;; Delta::CRY2/". Endogenous Delta tagged intracellularly with CRY2

w;; Delta::EGFP::CRY2/TM6B,Tb. Endogenous Delta tagged intracellularly with EGFP and CRY2

w;; Delta::TAGRFP::CRY2/TM6B,Tb. Endogenous Delta tagged intracellularly with TAGRFP and CRY2

w;; Delta::TAGRFP::CRY2-olig/TM6B,Tb. Endogenous Delta tagged intracellularly with TAGRFP and CRY2-olig (increased clustering variant of CRY2)

w;; Delta::CIBN/". Endogenous Delta tagged intracellularly with CRY2 dimerizing partner CIBN.

Notch-Extra::YFP/Y (DGRC-Kyoto Line 115544). Endogenous Notch tagged extracellularly with YFP at amino acid 55.

Notch-Extra::YFP/Y;; Delta::CRY2/"

yw; P[w+, nosP > MCP-no nls::GFP]; +/+ (from Thomas Gregor)

w; P[w+, nosP > MCP-no nls::GFP]/CyO; MKRS/TM3,Ser

w; P[w+, nosP > MCP-no nls::GFP]/CyO; Delta::CRY2/"

w;; Sim-MS2/"

w;; FRT82B Delta::CRY2/"

w; Ubx-FLP neur-iRFP670nls/CyOGFP; FRT82B ubi-RFPnls [42]

yw; EGFP::Rab5/CyO [46]

yw FRT19A Notch[55e11]/FM7c, Notch loss-of-function mutation (Eric Wieschaus)

w;; Df(3R)Dl-KX23, e[*]/TM3, Ser (BL-2411), Chromosomal deficiency in 3R lacking Delta

w;; Df(3R)BSC850/TM6C, Sb cu (BL-27922), Chromosomal deficiency in 3R lacking Delta

**Expanded View** for this article is available online.

## Acknowledgements

We thank all members of the De Renzis laboratory for helpful discussion. We thank C. Tischer for helping with image analysis and the data pipeline developed in Cell Profiler for quantifying Delta clustering kinetics. We thank A. Aulehla, J. Crocker, J. Hartmann and T. Hiiragi for helpful discussions and critical reading of the manuscript. We thank the advanced light microscopy for their advice and assistance. This work was supported by EMBL internal funding available to S.D.R. We thank the Bloomington *Drosophila* Stock Center and the *Drosophila* Genomics Resource Center for providing fly stocks and cDNAs. We thank M. Levine for providing fly stocks.

## Author contributions

The experiments were conceived and designed by RV, AN, PN and SDR. AN generated the endogenously tagged Delta lines and collected the data presented in Figs 1D–G and P, Q and 2B and C. RV collected all the other data with help of MT and FS for Fig 3N and O, and RKH for Fig 2B and C. DK analysed the data presented in Figs 3I and EV2 and helped with figure preparation. EE generated the sim-MS2 reporter line, and PN analysed the data presented in Fig 4D–H and helped with data analysis and figure preparation. AN and RV wrote the manuscript with input from SDR, PN and FS.

## Conflict of interest

The authors declare that they have no conflict of interest.

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
