## [Review Process File · EMBO Reports]

Optogenetic inhibition of Delta reveals digital Notch signaling output during tissue differentiation

Ranjith Viswanathan, Aleksandar Necakov, Mateusz Trylinski, Rohit Krishnan Harish, Daniel Krueger, Emilia Esposito, Francois Schweisguth, Pierre Neveu, and Stefano De Renzis

Review timeline:

Submission date:	25 February 2019
Editorial Decision:	2 April 2019
Revision received:	16 August 2019
Editorial Decision:	19 September 2019
Revision received:	26 September 2019
Accepted:	4 October 2019

Editor: Deniz Senyilmaz-Tiebe

Transaction Report:

1st Editorial Decision

2 April 2019

Thank you for submitting your manuscript for consideration by EMBO Reports. Three referees agreed to review your manuscript. So far, we have received two referee reports that are copied below. Given that both referees are in fair agreement that you should be given a chance to revise the manuscript, I would like to ask you to begin revising your study along the lines suggested by the referees.

Please note that this is a preliminary decision made in the interest of time, and that it is subject to change should the third referee offer very strong and convincing reasons for this. As soon as we will receive the final report on your manuscript, we will forward it to you as well.

As you can see, both referees express interest in the presented method for optogenetic control of Notch signaling and the proposed modes of Notch signaling during differentiation. However, they also raise concerns that need to be addressed in full before we can consider publication of the manuscript here. In particular, the referee #2 requires additional data supporting the proposed differential modes of Notch signaling at tissue and cell levels.

Given these constructive comments, I would like to invite you to revise your manuscript with the understanding that the referee concerns must be fully addressed and their suggestions taken on board. Please address all referee concerns in a complete point-by-point response. Acceptance of the manuscript will depend on a positive outcome of a second round of review. It is EMBO Reports policy to allow a single round of revision only and acceptance or rejection of the manuscript will therefore depend on the completeness of your responses included in the next, final version of the manuscript.

REFEREE REPORTS

Referee #1:

Viswanathan and colleagues investigate how Notch signals are processed during *Drosophila* embryonic development. They develop an endogenously-tagged optogenetic allele of Delta based on the CRY2/CIB1 protein dimerization system. Delta::CRY2 embryos, larvae and pupae grown in the presence of light developed Notch-like phenotypes, suggesting that tagging with CRY2, which induces oligomerization, is sufficient to inhibit Notch signaling. Upon photoactivation in the embryo, Delta::CRY2::GFP formed clusters that colocalized with Notch clusters, suggesting that Delta clusters can engage the receptor. Notably, photoactivation in the mesoderm resulted in increased retention of the intracellular domain of Notch in the plasma membrane and reduced transcription of the Notch target single-minded (*sim*) in the mesoectoderm, indicating that, even though Notch and Delta::CRY2 colocalize, signalling may be disrupted. To investigate if the defect in signaling was in the sending or the receiving end, the authors turned to clonal analysis in the pupal notum, where they made clones of opto-Delta homozygous cells. In the notum, sensory organ precursors (SOPs) are specified when one cell sends a Notch signal and represses SOP fate in its neighbors that receive the signal. Photoactivation resulted in the formation of SOPs predominantly inside the clones, suggesting that opto-Delta cells cannot effectively receive the signal when photoactivated, possibly due to cis-processing defects, but can send the signal. Next, the authors use the MS2-MCP system to investigate the dynamics of *sim* transcription. *sim* transcription begins at the end of cellularization in some nuclei, and increases progressively until all mesoectoderm nuclei displayed *sim* transcription. To determine why *sim* transcription is not simultaneous in all nuclei, the authors used mathematical modelling that suggested that a critical threshold of signalling is necessary for Notch activation within individual cells. Using opto-Delta to inhibit Notch signaling at different times after cellularization, the authors found that the earlier the inhibition happened, the longer the delay in *sim* expression, with 20 min of signaling as the minimum necessary to achieve the activation rate observed in controls. In contrast to the tissue-level delay, *sim* nuclear dots were either absent or present in individual nuclei, with no changes in intensity regardless of the time that the Notch signal had been enabled. The authors propose that Notch signaling acts as a switch to induce *sim* expression in individual cells, and as a dial at the tissue level to control the timing of *sim* expression.

This is a beautiful study in which the authors developed a number of new tools and came to really interesting conclusions about how Notch signals are processed. The manuscript tells two stories, with the common theme of Notch signaling. In practice, however, the two stories seem a bit disconnected, and I would strongly encourage the authors to try to better link them in their narrative. I also have some concerns about data interpretation that I think should be addressed before further considering the manuscript.

MAJOR

1. Page 5: the authors argue that the Delta::GFP flies they generated are fully viable, and that the construct can rescue both a Delta loss-of-function mutant and a deficiency in *trans*. However, none of these results are documented with data (or a reference). The authors should show the data that demonstrates the functionality of the Delta::GFP construct.

2. Figure 2D-G: the authors claim that the dynamics of Delta internalization are not affected by CRY2-induced Delta-clustering. However, based on Figure 2D, F cell interfaces seem discontinuous in the ectoderm upon photoactivation, and more cytoplasmic puncta (maybe bigger ones?) are present in the mesoderm. The authors should quantify the ratio of membrane to cytoplasmic signal in both ectoderm and mesoderm to show that the dynamics of internalization are not affected by photoactivation, because based on the images that they show, I'd think that Delta is more frequently in the cytoplasm upon photoactivation, which could also be consistent with the increased retention of Notch at the plasma membrane (Figure 3I).

3. Related to the previous point, what is the nature of the Delta-containing cytoplasmic compartments in the presence/absence of photoactivation? Are they endosomes in both cases or is it possible that photoactivation-induced clustering triggers increased Delta degradation?

4. Figure EV1: the colocalization of Notch and Delta::CRY2 in clusters upon photoactivation suggests that Delta::CRY2 clusters can engage Notch. However, colocalization is not evidence for interaction. FRET would be significantly more convincing here. Particularly in light of the fact that

NICD is retained in the plasma membrane upon photoactivation.

MINOR

1. Page 6: why do Delta::CRY2::GFP, Delta::CRY2::RFP and Delta::CRY2-olig have problems to produce homozygous flies when raised in the dark, and Delta::CRY2 and Delta::CIBN do not? The authors should at least try to interpret this result.
2. Page 8, Figure 3C: the authors indicate that some endocytic NECD vesicles formed in the mesectoderm upon photoactivation. They should add arrowheads to the inset in Figure 3C to clearly indicate where are those vesicles. I think I know what they are referring to, but it is not totally clear. Furthermore, they may want to refer to these as vesicles, rather than "endocytic vesicles", as they have shown no evidence that these vesicles are indeed endosomes.

TYPOS

1. Page 8: "opto-Delta clusters in the ectoderm contained Notch (Fig. 1C)". The reference to Fig. 1C is wrong and should be corrected. Maybe Fig. EV1C?
2. Page 18: "identifies and segments" should be "identify and segment".
3. Page 11, paragraph 2, line 2: the equation is missing one parenthesis at the end.
4. Page 21, paragraph 2: the two equations are missing one parenthesis at the end each.

Referee #2:

The manuscript by Viswanathan et al. describes an opto-genetic method to control Notch signaling in *Drosophila* in vivo and use this tool to study the dynamic input-output relation between Notch activation and transcriptional response of a target gene. The first 3 figures of the manuscript describe the construction and characterization of light inducible Delta. They find that the light inducible Delta cis-inhibits Notch when light is turned on. The last figure presents an effort to use this tool to study the transcriptional response of a Notch target (Sim) to varying levels of Notch activation. While the first part is a remarkable technical achievement and experiments are quite comprehensive and convincing, the second part is disappointing and does not provide strong evidence for the claims in the title and abstract. The authors need to perform additional experiments that support their claims.

Specific comments:

1. Figure 4I shows that embryos in which Notch is inhibited at different time points exhibit delayed onset of transcription. The suggested interpretation is that different inhibitions correspond to different levels of Notch activation (e.g. NICD levels). However, the light induced inactivation of Notch does not only change the level of Notch, but also changes the timing of activation. This is not taken into account in their suggested model. The authors should verify that the level of Notch is what matters by varying this level in other methods, such as changing the copy number of Notch.
2. The statistics use in the experiments for Fig. 4 are rather low, and some data points are missing in some panels. This includes:
 - a. Why does the 15min photoactivation is missing in Fig. 4D-K?
 - b. Why panel J is lacking the 25min time point (shown in I)?
 - c. Why does panel L lacks the 15,20,25 min time points?
 Furthermore, it seems that most of the data shown in Fig. 4 is based on analysis of 3 embryos per condition (it says $N \geq 3$ in caption). Given the stochastic nature of the system I believe that the authors should provide more statistics to support their claims.
3. The model is quite unclear and not fully justified. The assumption for the model is that different Notch activity level (NICD levels) control the parameter "k" referred to as the activation rate. This is unclear, since in standard models of transcription activation, the level of Notch should affect the level of NICD and that should affect the rate of transcription and the steady state (e.g. S_{max} in the current model). The time scale should be determined by the lifetime of the mRNA in the spot. Hence, the assumptions of the model should be clarified before conclusions can be drawn from it. Furthermore, the use of different parameters in different conditions is confusing (S_{max} vs N_{max} , k vs K).
4. The authors claim that at the cell levels the Notch activation shows a switch-like behavior. This is

based on Fig. 4L that shows that the spot intensity is constant. First, as mentioned before, the data seems to be lacking other time points. Second, the authors should provide additional evidence for the switch like behavior (especially since the use of the MS2 reporter is performed for the first time in this context). This can be done by performing mRNA labeling in fixed samples and by using existing Notch transcriptional reporters. This is essential in order to substantiate these results.

5. The correlation in Fig. 4J should be provided with a p-value to confirm its statistical significance.

6. Even if the conclusions are correct, the physiological implications are unclear. There is no discussion anywhere regarding the functional implications of the proposed mechanism.

7. Regarding Figures 2B and 2H,I. How come the clustering in Notch is observed after only 5sec while the clustering in Delta is observed after 120 sec? what happens at comparable times?

Additional Correspondence

26 April 2019

We have in the mean time received the report from referee #3, who also finds the study interesting and in principle supports publication. As I mentioned in my previous letter, please address all comments of this referee as well during revision.

I am looking forward to reading your revised manuscript when it is ready.

Referee #3

This paper by De Renzis and colleagues expands the optogenetic toolbox with a light-controlled Delta (opto-Delta). Like recent work in *Drosophila* from the Saunders group (Opto-Bicoid in *eLife*) and Harrison group (Opto-Zelda in *Mol Cell*), this opto-Delta ligand is inactivated by light-induced clustering of a Cry2 fusion protein, functioning as a light-inducible loss of function. Like each of these previous tools, it is not obvious how clustering results in the loss-of-function phenotype, and the authors do a nice job of digging into this question (cis vs trans-repression).

In general I am very enthusiastic about this paper, which has high novelty (the first light-controlled Notch/Delta system) and which sheds some light onto the mechanism of Cry2-based inhibition. I only have a few suggestions to improve the paper:

- Include at least some comparisons of light-treated and classic genetic loss-of-function phenotypes to benchmark the inhibition. For a *Drosophilist* or Notch aficionado, it is probably obvious how the loss of function results in Fig. 1 and Fig. 2J-K compare to loss-of-function mutants with different strengths. But for a journal with the broad readership of EMBO J (which is not limited to developmental biologists) it is difficult to immediately understand how potent the light-induced loss-of-function activity is, and which mutants make the best comparison. Does light inhibit Notch by 50%? 90%? 99%? At least in some of these cases, a better comparison to benchmarks where the level of inhibition is known would be extremely helpful (e.g. hemizygous or complete loss-of-function mutants).

This is crucial because anyone who would like to use this tool should be able to know the total dynamic range of this tool - how much activity switches from light to dark.

- The "stochastic model" of Figure 4 is not a stochastic model - the curves are perfectly deterministic, and represent the solution of a first-order mass-action chemical reaction (an exponential approach to a new steady-state after reaction conditions are changed). The concepts and results that are presented are equally true of any biochemical system with large #s of molecules, where stochastic, noisy effects can be neglected. The authors should just change the terminology they use to describe the model (perhaps "simulation of a first-order chemical reaction" vs "first-order chemical reaction where output only occurs above a threshold").

- My last comment is about beautiful recent work from the Elowitz lab (Nandagopal et al, *Cell* 2018) that indicates Notch signaling can be either transient or sustained, and that suggested clustering of Delta is the determinant of whether Notch dynamics are pulsatile or sustained. I know it is a very different system (mammalian vs *Drosophila*, with multiple Dll ligands) but could transient vs sustained Notch signaling result from optogenetic clustering induced here? Is this a testable hypothesis in the fly?

Referee #1:

Viswanathan and colleagues investigate how Notch signals are processed during *Drosophila* embryonic development. They develop an endogenously-tagged optogenetic allele of Delta based on the CRY2/CIB1 protein dimerization system. Delta::CRY2 embryos, larvae and pupae grown in the presence of light developed Notch-like phenotypes, suggesting that tagging with CRY2, which induces oligomerization, is sufficient to inhibit Notch signaling. Upon photoactivation in the embryo, Delta::CRY2::GFP formed clusters that colocalized with Notch clusters, suggesting that Delta clusters can engage the receptor. Notably, photoactivation in the mesoderm resulted in increased retention of the intracellular domain of Notch in the plasma membrane and reduced transcription of the Notch target single-minded (*sim*) in the mesoectoderm, indicating that, even though Notch and Delta::CRY2 colocalize, signalling may be disrupted. To investigate if the defect in signaling was in the sending or the receiving end, the authors turned to clonal analysis in the pupal notum, where they made clones of opto-Delta homozygous cells. In the notum, sensory organ precursors (SOPs) are specified when one cell sends a Notch signal and represses SOP fate in its neighbors that receive the signal. Photoactivation resulted in the formation of SOPs predominantly inside the clones, suggesting that opto-Delta cells cannot effectively receive the signal when photoactivated, possibly due to cis-processing defects, but can send the signal. Next, the authors use the MS2-MCP system to investigate the dynamics of *sim* transcription. *sim* transcription begins at the end of cellularization in some nuclei, and increases progressively until all mesoectoderm nuclei displayed *sim* transcription. To determine why *sim* transcription is not simultaneous in all nuclei, the authors used mathematical modelling that suggested that a critical threshold of signalling is necessary for Notch activation within individual cells. Using opto-Delta to inhibit Notch signaling at different times after cellularization, the authors found that the earlier the inhibition happened, the longer the delay in *sim* expression, with 20 min of signaling as the minimum necessary to achieve the activation rate observed in controls. In contrast to the tissue-level delay, *sim* nuclear dots were either absent or present in individual nuclei, with no changes in intensity regardless of the time that the Notch signal had been enabled. The authors propose that Notch signaling acts as a switch to induce *sim* expression in individual cells, and as a dial at the tissue level to control the timing of *sim* expression.

This is a beautiful study in which the authors developed a number of new tools and came to really interesting conclusions about how Notch signals are processed. The manuscript tells two stories, with the common theme of Notch signaling. In practice, however, the two stories seem a bit disconnected, and I would strongly encourage the authors to try to better link them in their narrative. I also have some concerns about data interpretation that I think should be addressed before further considering the manuscript.

We have expanded the introduction to better link the two parts of the manuscript and provide more context on the questions that have been addressed in the paper, with particular emphasis on the relationship between the dynamics of Notch signalling activation and target gene expression. We have also revised the result and discussion section accordingly.

MAJOR

1. Page 5: the authors argue that the Delta::GFP flies they generated are fully viable, and that the construct can rescue both a Delta loss-of-function mutant and a deficiency in trans. However, none of these results are documented with data (or a reference). The authors should show the data that demonstrates the functionality of the Delta::GFP construct.

These data have been discussed in the text and presented in Fig. EV1A-C.

2. Figure 2D-G: the authors claim that the dynamics of Delta internalization are not affected by CRY2-induced Delta-clustering. However, based on Figure 2D, F cell interfaces seem discontinuous in the ectoderm upon photoactivation, and more cytoplasmic puncta (maybe bigger ones?) are

present in the mesoderm. The authors should quantify the ratio of membrane to cytoplasmic signal in both ectoderm and mesoderm to show that the dynamics of internalization are not affected by photoactivation, because based on the images that they show, I'd think that Delta is more frequently in the cytoplasm upon photoactivation, which could also be consistent with the increased retention of Notch at the plasma membrane (Figure 3I).

We have quantified Delta plasma membrane levels in the mesoderm and ectoderm. These data are discussed in the text (p.9, lanes 5-13) and presented in Fig. EV2A-D. These results show that Delta internalization does not significantly change upon light activation. Fig. EV2A-C shows that the plasma membrane levels of Delta in the ectoderm are 2-3 times higher than in the mesoderm both in the dark and light condition. Fig. 2EVD shows that in the mesoderm the plasma membrane to cytoplasmic (vesicles) ratio is ~0.3 and this ratio does not significantly change upon light exposure. We have also updated panels D-F in Figure 2 to show the exactly corresponding focal planes between the dark and light conditions.

3. Related to the previous point, what is the nature of the Delta-containing cytoplasmic compartments in the presence/absence of photoactivation? Are they endosomes in both cases or is it possible that photoactivation-induced clustering triggers increased Delta degradation?

Delta vesicles correspond to endosomes as demonstrated by co-localization with the early endosomal marker Rab5 (Fig. 2VE-H). About 75% of Delta vesicles are positive for Rab5 both in light and dark condition (Fig. 2EVH) arguing against the possibility that photo-activation is triggering a different trafficking route leading to increased degradation.

4. Figure EV1: the colocalization of Notch and Delta::CRY2 in clusters upon photoactivation suggests that Delta::CRY2 clusters can engage Notch. However, colocalization is not evidence for interaction. FRET would be significantly more convincing here. Particularly in light of the fact that NICD is retained in the plasma membrane upon photoactivation.

We agree that co-localization does not prove interaction and we have specifically stated this in the text (p.9, lanes 17-19) **Antibody staining demonstrated Notch and Delta colocalization in ectodermal clusters, showing that opto-Delta clusters caused co-clustering of Notch and presumably its engagement (Fig. EV3B-D).** We do not have the fly lines to perform FRET measurements and it would take a significant amount of time to generate new lines.

MINOR

1. Page 6: why do Delta::CRY2::GFP, Delta::CRY2::RFP and Delta::CRY2-olig have problems to produce homozygous flies when raised in the dark, and Delta::CRY2 and Delta::CIBN do not? The authors should at least try to interpret this result.

We do not know why this is the case, but we have added the following explanation (p.7, lanes 12-15) **This might be caused either by the concomitant presence of two relatively large tags (each ~ 35 kDa), which could interfere with protein folding, or by increased dark activity (i.e. the excited state of a photoreceptor in the dark) of the CRY2-olig tag compared to CRY2.**

2. Page 8, Figure 3C: the authors indicate that some endocytic NECD vesicles formed in the mesectoderm upon photoactivation. They should add arrowheads to the inset in Figure 3C to clearly indicate where are those vesicles. I think I know what they are referring to, but it is not totally clear. Furthermore, they may want to refer to these as vesicles, rather than "endocytic vesicles", as they have shown no evidence that these vesicles are indeed endosomes.

We have added arrows to the panel and removed endosomes from NECD —although we have previously shown that NECD vesicles are endosomes (De Renzis Dev Cell 2006 and Bardin A Dev Cell 2006)

TYPOS

1. Page 8: "opto-Delta clusters in the ectoderm contained Notch (Fig. 1C)". The reference to Fig. 1C is wrong and should be corrected. Maybe Fig. EV1C?

Corrected to Fig. EV3B-D

2. Page 18: "identifies and segments" should be "identify and segment".

Corrected

3. Page 11, paragraph 2, line 2: the equation is missing one parenthesis at the end.

4. Page 21, paragraph 2: the two equations are missing one parenthesis at the end each.

In this revised version of the manuscript we did not include the model as we felt (also by taking into account reviewer #2 and #3 comments) it did not really help explaining our results.

Referee #2:

The manuscript by Viswanathan et al. describes an opto-genetic method to control Notch signaling in *Drosophila* in vivo and use this tool to study the dynamic input-output relation between Notch activation and transcriptional response of a target gene. The first 3 figures of the manuscript describe the construction and characterization of light inducible Delta. They find that the light inducible Delta cis-inhibits Notch when light is turned on. The last figure presents an effort to use this tool to study the transcriptional response of a Notch target (*Sim*) to varying levels of Notch activation. While the first part is a remarkable technical achievement and experiments are quite comprehensive and convincing, the second part is disappointing and does not provide strong evidence for the claims in the title and abstract. The authors need to perform additional experiments that support their claims.

Specific comments:

1. Figure 4I shows that embryos in which Notch is inhibited at different time points exhibit delayed onset of transcription. The suggested interpretation is that different inhibitions correspond to different levels of Notch activation (e.g. NICD levels). However, the light induced inactivation of Notch does not only change the level of Notch, but also changes the timing of activation. This is not taken into account in their suggested model. The authors should verify that the level of Notch is what matters by varying this level in other methods, such as changing the copy number of Notch.

We thank this reviewer for suggesting this experiment which has helped to strengthen our conclusion. In the new Fig. EV5 B,C we have analysed the dynamics of *sim* expression in Notch heterozygous embryo using the MS2/MCP system. In agreement with the hypothesis that Notch levels control the timing of activation, the result of this experiments shows that in Notch heterozygous embryo *sim* transcription is delayed of about 5 min.

2. The statistics use in the experiments for Fig. 4 are rather low, and some data points are missing in some panels. This includes:

We have now increased the number of replicates to at least 5 in each experimental condition. The exact numbers are reported in the corresponding legends. Furthermore we have removed the model as we felt it did not really help explaining our data and we have rearranged the entire figure 4 to include the new data.

a. Why does the 15min photoactivation is missing in Fig. 4D-K?

We have now performed the 15 min timepoints and the new data have been added in Fig.4E-H

b. Why panel J is lacking the 25min time point (shown in I)?

In the new Fig. 4 we present the activation rates and onset of *sim* transcription as separate bar graphs and both include the 25 min time points (Fig. 4F,G) and therefore Panel J is no longer displayed.

c. Why does panel L lacks the 15,20,25 min time points?

This panel is now panel H and all the time-points have been included.

Furthermore, it seems that most of the data shown in Fig. 4 is based on analysis of 3 embryos per condition (it says $N \geq 3$ in caption). Given the stochastic nature of the system I believe that the authors should provide more statistics to support their claims.

Now we have analysed at least 5 embryos per condition.

3. The model is quite unclear and not fully justified. The assumption for the model is that different Notch activity level (NICD levels) control the parameter "k" referred to as the activation rate. This is unclear, since in standard models of transcription activation, the level of Notch should affect the level of NICD and that should affect the rate of transcription and the steady state (e.g. S_{max} in the current model). The time scale should be determined by the lifetime of the mRNA in the spot. Hence, the assumptions of the model should be clarified before conclusions can be drawn from it. Furthermore, the use of different parameters in different conditions is confusing (S_{max} vs N_{max} , k vs K).

We agree that the model did not really help explain our results and therefore decided to remove it.

4. The authors claim that at the cell levels the Notch activation shows a switch-like behavior. This is based on Fig. 4L that shows that the spot intensity is constant. First, as mentioned before, the data seems to be lacking other time points. Second, the authors should provide additional evidence for the switch like behavior (especially since the use of the MS2 reporter is performed for the first time in this context). This can be done by performing mRNA labeling in fixed samples and by using existing Notch transcriptional reporters. This is essential in order to substantiate these results.

We have performed additional experiments to back up this conclusion using quantitative fluorescent in situ hybridization of nascent transcripts. These data are presented in Fig. EV5D-I and demonstrate no increase in sim spot intensity over time, there was only a 20% variation in spot intensity over time, a value that was smaller than the ~75% variation in spot intensity for a given time point.

5. The correlation in Fig. 4J should be provided with a p-value to confirm its statistical significance.

This panel is no longer displayed.

6. Even if the conclusions are correct, the physiological implications are unclear. There is no discussion anywhere regarding the functional implications of the proposed mechanism.

We have now discussed the functional implication of our results (p14, lanes 13-19)

“The physiological implications of this regulatory mode can be understood in the context of thresholded non-linear responses, which are known to confer robustness of signalling outputs to fluctuation in inputs {Mc Mahon, 2014 #2025}. It is tempting to speculate that time-integrated analog-to-digital conversion of Notch signalling may serve the function to minimize spurious target gene expression resulting from transient cell-cell contacts during highly dynamics morphogenetic movements”.

7. Regarding Figures 2B and 2H,I. How come the clustering in Notch is observed after only 5sec while the clustering in Delta is observed after 120 sec? what happens at comparable times?

This is because the Notch clustering experiments were done in a Delta::CRY2 homozygous condition. Delta::CRY2 homozygous cluster much faster than Delta::GFP::CRY2 heterozygous. We have now repeated Notch clustering in a Delta::CRY2 heterozygous background demonstrating similar kinetics of clustering as Delta::GFP::CRY2 heterozygous (120 s). Notch clustering under Delta::CRY2 homozygous condition is presented in Fig. EV3A.

Referee #3

This paper by De Renzis and colleagues expands the optogenetic toolbox with a light-controlled Delta (opto-Delta). Like recent work in Drosophila from the Saunders group (Opto-Bicoid in eLife)

and Harrison group (Opto-Zelda in Mol Cell), this opto-Delta ligand is inactivated by light-induced clustering of a Cry2 fusion protein, functioning as a light-inducible loss of function. Like each of these previous tools, it is not obvious how clustering results in the loss-of-function phenotype, and the authors do a nice job of digging into this question (cis vs trans-repression).

In general I am very enthusiastic about this paper, which has high novelty (the first light-controlled Notch/Delta system) and which sheds some light onto the mechanism of Cry2-based inhibition. I only have a few suggestions to improve the paper:

- Include at least some comparisons of light-treated and classic genetic loss-of-function phenotypes to benchmark the inhibition. For a Drosophilist or Notch aficionado, it is probably obvious how the loss of function results in Fig. 1 and Fig. 2J-K compare to loss-of-function mutants with different strengths. But for a journal with the broad readership of EMBO J (which is not limited to developmental biologists) it is difficult to immediately understand how potent the light-induced loss-of-function activity is, and which mutants make the best comparison. Does light inhibit Notch by 50%? 90%? 99%? At least in some of these cases, a better comparison to benchmarks where the level of inhibition is known would be extremely helpful (e.g. hemizygous or complete loss-of-function mutants). This is crucial because anyone who would like to use this tool should be able to know the total dynamic range of this tool - how much activity switches from light to dark.

We have added new Fig. EV1D-L where we compare cuticle, eye, and wing phenotypes of Delta heterozygous and Delta::CRY2 homozygous flies exposed to light. These results argue that optogenetic activation leads to more than 50% inhibition of Delta activity.

- The "stochastic model" of Figure 4 is not a stochastic model - the curves are perfectly deterministic, and represent the solution of a first-order mass-action chemical reaction (an exponential approach to a new steady-state after reaction conditions are changed). The concepts and results that are presented are equally true of any biochemical system with large #s of molecules, where stochastic, noisy effects can be neglected. The authors should just change the terminology they use to describe the model (perhaps "simulation of a first-order chemical reaction" vs "first-order chemical reaction where output only occurs above a threshold").

We have decided not to include the model as based also on reviewer 2 feedback
We felt that it did not really help explain our results.

- My last comment is about beautiful recent work from the Elowitz lab (Nandagopal et al, Cell 2018) that indicates Notch signaling can be either transient or sustained, and that suggested clustering of Delta is the determinant of whether Notch dynamics are pulsatile or sustained. I know it is a very different system (mammalian vs Drosophila, with multiple Dll ligands) but could transient vs sustained Notch signaling result from optogenetic clustering induced here? Is this a testable hypothesis in the fly?

We have cited this paper in the introduction as we agree it provides a nice example of dynamic regulation of Notch signalling. However, we do not think that the optogenetic clustering we induced in our system can be easily related to transient vs sustained activation mainly because opto-Delta cluster inhibit signalling.

2nd Editorial Decision

19 September 2019

Thank you for submitting your revised manuscript. It has now been seen by all of the original referees.

As you can see, the referees find that the study is significantly improved during revision and recommend publication. Before I can accept the manuscript, I need you to address some editorial points below:

- Please address the remaining minor concerns of referee #2.

REFeree REPORTS**Referee #1:**

This is a well-executed revision. The authors have addressed most of my concerns. I support publication.

Referee #2:

The revised manuscript by Viswanathan et al. is significantly improved compared to the first version. The additional analysis of Notch heterozygous mutant and the control for the MS2-MCP using smFISH are important additions to the manuscript and strengthen its conclusions. I also agree with the decision to remove the model as the conclusions are clear without it. I therefore recommend accepting the manuscript for publication.

minor comments:

1. The manuscript mentions the two very recent papers of Falo-San Juan and Lammers addressing similar questions. I suggest also addressing these in the discussion to highlight similarities/differences in the conclusions.
2. The 15min time point is still missing in Fig EV5a
3. There is a typo in the y-axis title of Fig. EV5c (onst instead of Onset).

Referee #3:

The authors have addressed all of my concerns. I congratulate them on a very nice paper!

2nd Revision - authors' response

26 September 2019

Referee #2:

The revised manuscript by Viswanathan et al. is significantly improved compared to the first version. The additional analysis of Notch heterozygous mutant and the control for the MS2-MCP using smFISH are important additions to the manuscript and strengthen its conclusions. I also agree with the decision to remove the model as the conclusions are clear without it. I therefore recommend accepting the manuscript for publication.

minor comments:

1. The manuscript mentions the two very recent papers of Falo-San Juan and Lammers addressing similar questions. I suggest also addressing these in the discussion to highlight similarities/differences in the conclusions.

We have cited both papers in the introduction and have already highlighted in our previous version of the manuscript the similarities of our results in the result section when relevant. We have decided not to discuss further these two studies as this would require a detailed explanation of transcriptional bursting, which is something that we have not addressed in our study.

2. The 15min time point is still missing in Fig EV5a

Done

3. There is a typo in the y-axis title of Fig. EV5c (onst instead of Onset).

Done.

3rd Editorial Decision

04 October 2019

Thank you for submitting your revised manuscript. I have now looked at everything and all looks fine. Therefore I am very pleased to accept your manuscript for publication in EMBO Reports.

Corresponding Author Name: Stefano De Renzi

Manuscript Number: EMBOR-2019-47999V1